# Delayed Surgical Treatment of Displaced Intra-Articular Calcaneal Fractures in Major Trauma Is Safe and Effective

**DOI:** 10.3390/jcm12052039

**Published:** 2023-03-04

**Authors:** Attilio Basile, Riccardo Maria Lanzetti, Alessio Giai Via, Teresa Venditto, Marco Spoliti, Pasquale Sessa, Mauro Tortora, Nicola Maffulli

**Affiliations:** 1Orthopaedic and Traumatology Unit, Department Emergency and Acceptance, San Camillo-Forlanini Hospital, 00153 Rome, Italy; 2Department of Primary Care, Rehabilitation and Prosthetic, AUSL Latina, 04100 Latina, Italy; 3Department of Musculoskeletal Disorder, Faculty of Medicine and Surgery, University of Salerno, 84084 Salerno, Italy; 4Centre for Sports and Exercise Medicine, Barts and the London School of Medicine and Dentistry, Mile End Hospital, Queen Mary University of London, 275 Bancroft Road, London E1 4DG, UK

**Keywords:** calcaneus fracture, sinus tarsi approach, complication, major trauma, polytrauma patients, delayed surgery

## Abstract

Background: To assess whether delaying operative fixation through the sinus tarsi approach resulted in a decreased wound complications rate or could hinder the quality of reduction in subjects with Sanders type II and III displaced intra-articular calcaneus fractures. Methods: From January 2015 to December 2019, all polytrauma patients were screened for eligibility. We divided patients into two groups: Group A, treated within 21 days after injury; Group B, treated more than 21 days after injury. Wound infections were recorded. Radiographic assessment consisted of serial radiographs and CT scans: postoperatively (T0) and at 12 weeks (T1) and at 12 months after surgery (T2). The quality of reduction of the posterior subtalar joint facet and calcaneal cuboid joint (CCJ) was classified as anatomical and non-anatomical. A post hoc power calculation was performed. Results: A total of 54 subjects were enrolled. Four wound complications (three superficial, one deep) were identified in Group A; two wound complications (one superficial one deep) were identified in Group B. According to “mean interval between trauma and surgery” and “duration of intervention”, there was a significant difference between the groups (*p* < 0.001). There were no significant differences between Groups A and B in terms of wound complications or quality of reduction. Conclusions: The sinus tarsi approach is a valuable approach for the surgical treatment of closed displaced intra-articular calcaneus fractures in major trauma patients who need delayed surgery. The timing of surgery did not negatively influence the quality of the reduction and the wound complication rate. Level of evidence: level II, prospective comparative study.

## 1. Introduction

Calcaneal fractures are common, about 1–2% of all fractures, and they are mostly due to high-energy injuries that occur in young, active, laboring individuals, with a consequently high negative socioeconomic impact [1,2]. Approximately 70% of calcaneal fractures are displaced intra-articular calcaneal fractures (DIACFs) [3,4].

The goal of management is accurate anatomic reduction and stable fixation for early functional rehabilitation avoiding soft tissue complications [5]. Open reduction and internal fixation (ORIF) with the extensile lateral approach (ELA) is considered the gold standard treatment of these fractures [6,7]. This approach provides excellent exposure of the posterior facet and lateral wall. However, it has been associated with high rates of wound complication and infection [8,9]. Minimal incisions, including the sinus tarsi approach (STA), have been introduced to reduce the risk of wound complications [7]. However, the STA allows only a restricted view of fracture area, and is technically demanding. The results are comparable to the ELA, and it might be a valuable resource in the treatment decision making [10].

The optimal timing of surgical stabilization of calcaneal fractures is controversial. Surgeon’s experience, fracture severity, soft tissue impairment, and surgical timing are the main risk factors for complications [11]. Poorer outcomes have also been reported for polytrauma and psychiatric patients [12]. The incidence of deep infections after the ELA varies widely, ranging from 1.8% to 21% [13]. Early fixation of these fractures has been associated with skin flap necrosis, infection, and the need for multiple surgical interventions [14]. To decrease the rate of wound complications, patients may benefit from delayed surgery, considering that 7 to 14 days are often required for the swelling to settle [15]. However, prolonged surgical delay may increase postoperative infection rates and the risks of suboptimal reduction [16]. Therefore, the optimal timing to operative fixation using the sinus tarsi approach has not been determined [17].

This observational clinical study assessed whether delayed ORIF of displaced calcaneus fractures through the sinus tarsi approach (3 weeks or longer after the trauma) resulted in decreased wound complication rates. We also assessed the effect of timing on quality of reduction. The null hypothesis is that no differences in both the complication rate and the quality of reduction could be detected between patients treated earlier than 21 days or later.

## 2. Materials and Methods

### 2.1. Study Design and Participants

This study was designed as prospective comparative study. From January 2015 to December 2019, all major trauma patients who were referred to the San Camillo-Forlanini Hospital Trauma Center of Rome (Italy) with a diagnosis of Displaced Intra-Articular Calcaneus Fractures were screened for eligibility.

The included subjects with calcaneal fracture were treated with the STA and internal fixation performed by a single experienced surgeon.

The study was conducted in accordance with the principles of the Declaration of Helsinki and its amendments. Patients were fully informed of the purposes of the study before providing their consent. The study was approved by the local ethical committee board (Comitato Etico Lazio 1, n. 0074/2019, 25 September 2019).

The inclusion criteria for this study were (1) displaced intra-articular fractures of the calcaneus with standard radiographic and CT evidence of Sanders type II or III fractures, (2) patients aged between 18 and 65, and (3) unilateral or bilateral calcaneal fractures.

Patients with an open fracture, peripheral vascular disease, skin infection, signs of compartment syndrome, neurological deficit following head injury or spinal injury, other fractures of the ipsilateral or contralateral limb, and severe osteoporosis were excluded. We excluded patients with previous major underlying medical comorbidities before the index traumatic event: uncontrolled hypertension, previous myocardial infarction, cancer, history of stroke or transient ischemic attacks, chronic obstructive lung disease, morbid obesity (Body Mass Index—BMI > 35), diabetes mellitus, peripheral vascular diseases, or peripheral neuropathies.

Patient’s characteristics (i.e., gender, age at trauma, BMI, comorbidities), injury characteristics (date of trauma, affected side, trauma mechanism, fracture classification, radiographic measurements, soft tissue compromise, complications) were obtained from the electronic patient’s medical files.

According to the interval between trauma and surgery, we divided patients into two groups:

Group A: patients treated within 21 days after injury;

Group B: patients treated more than 21 days after injury;

According to the US Centers for Disease Control and Prevention [18], wound infections were classified as:

“Superficial wound infections”: wound dehiscence or signs of infections (confirmed with a positive culture) amenable to conservative treatment with antibiotics.

“Deep wound infections”: osteomyelitis, infected hardware or a plate fistula needing hardware removal, (readmission with) intravenous antibiotics or wound debridement with or without local antibiotic treatment with gentamicin beads or vacuum-assisted closure. Infections were confirmed with a positive culture.

Calcaneus fractures were classified according to the Sanders classification (Table 1).

Follow-up radiographic assessment consisted of serial radiographs and CT scans. Radiographs included the lateral, 30° Broden, and Harris views taken postoperatively (T0) and at 12 weeks (T1) (no-weight-bearing). At 12 months after surgery (T2), weight-bearing lateral and Saltzman views were taken. CT scans were obtained using 1 mm cuts in three planes and were taken preoperatively to assess the fracture characteristics, postoperatively (T0) and 12 (T2) months after surgery. The quality of reduction of the posterior subtalar joint (PSJ) facet and calcaneal cuboid joint (CCJ) was classified as anatomical (joint surface step-off < 2 mm) and non-anatomical (joint surface step-off > 2 mm) in accordance with the method described by Janzen on the postoperative CT scan [19]. The Böhler angle was measured on the lateral standard X-Ray view pre- and postoperatively. Postoperative CT scans and X-rays were evaluated by independent surgeons (TV, MT, and PS), and the patient group was blinded to the investigators.

### 2.2. Surgical Procedure

Patients were placed in the lateral position; an ankle tourniquet was used for a maximum time of 2 h then deflated even if surgery has not ended (Table 2). A conventional C-arm image intensifier was used to evaluate the quality of the reduction intraoperatively. An incision was made on a line from the tip of the lateral malleolus to the base of the fourth metatarsal (Figure 1). For the fractures without involvement of the CCJ, the posterior facet was approached through a 5 cm incision over the sinus tarsi. Release of the inferior peroneal retinaculum was performed, and the peroneal tendons were retracted inferiorly. After transection of the calcaneofibular ligament (CFL) and the subtalar ligaments, the subtalar joint was opened. For the fractures with involvement of the CCJ, a longer skin incision was made (8 to 10 cm) (extended subtalar approach: ESTA) and the extensor digitorum brevis muscle was elevated as far as the CCJ. As the first step, the posterior tuberosity fragment was reduced to the sustentacular constant fragment.

A periosteal elevator was inserted in the main fracture line (between the sustentacular fragment and the posterior tuberosity) to directly disimpact and separate the fragments. Then, a 5 mm Schanz pin was introduced percutaneously (in a posterior to anterior direction) in the posterior tuberosity fragment and traction was applied manually along the axis of the calcaneus to correct the deformity. Frequently, a small periosteal elevator was used to lever the fragments. Sometimes, a medially based pin distractor (Hintermann distractor) was used to aid the reduction of the posterior tuberosity. After the reduction and provisional fixation with Kirschner wires (K-wires) of the posterior tuberosity to the sustentacular “constant” portion (Figure 2), we proceeded with reduction of the fragments from posterior to anterior. Direct reduction of the articular fragments was performed (posterior facet, CCJ). Specifically designed plates for the STA (Wave Plate, Tornier, Macroom, Ireland) using both locking and conventional screws, often combined with free screws were used for definitive fixation. No bone graft was used. Reduction of the fracture, restoration of the Bohler and Gissane angles, and adequate fixation were confirmed by intraoperative fluoroscopic lateral, axial, and Broden’s views. The subcutaneous layer was closed over a drain with the use of an inverted buried 2-0 absorbable suture. The skin was closed with interrupted horizontal Donati sutures. The drain was removed after 24 h.

### 2.3. Statistical Analysis

Statistical analysis was performed according to the principle of intention-to-treat, with missing data imputed with the ‘last observation carried forward’ technique. After using a Kolmogorov–Smirnov test to verify that the variables were normally distributed, we applied parametric tests.

Unpaired Student’s *t*-test for continuous data was used to compare the results between patient groups.

The χ^2^ statistic was used to test the association intervals between “trauma and surgery” and “duration of intervention”, “quality of reduction”, and “Sanders Classification”.

Social Sciences (SPSS) version 18 was used for calculations (IBM, Armonk, New York, NY, USA). All data were analyzed by an individual researcher. *p* values < 0.05 were considered statistically significant, and all results are expressed with a 95% confidence interval (CI).

### 2.4. Sample Size

Power calculation was computed assuming an effect size value of 0.4, a two-tailed value of 0.05 (sensitivity 95%), and a β value of 0.05 (study power: 95%); we determined that at least 54 patients were required (G Power3 power analysis program, Düsseldorf, Germany).

## 3. Results

A total of 54 subjects (mean age ± SD, 38.35 ± 11.77 years; range, 18–65 years; 22 women and 32 men) were enrolled in the study. Of the sample, 63% (N = 34) was enrolled in Group A and 37% (N = 20) in Group B, according to the interval between trauma and surgery (Figure 3). Five (N = 5) and three (N = 3) patients sustained bilateral calcaneal fractures in Groups A and B, respectively. Baseline characteristics of the sample are shown in Table 3.

Non-anatomical reduction was found in 3 of 34 (11%) and in 2 of 20 (10%) calcaneal fractures in groups A and B, respectively (χ^2^ = 1.24, *p* = 0.26). The χ^2^ test demonstrated no significant relationship between Groups A and B’s Sanders Classification (χ^2^ = 0.17, *p* = 0.56) and quality of reduction (χ^2^ = 0.01, *p* = 0.63) (Figure 4). The preoperative median Böhler angle was 6° (±5°) in group A and 4° (±6°) in group B (*p* = 0.45), and the postoperative median Böhler angle was 31° (±5°) and 29° (±6°), respectively, but the difference was not statistically significant (*p* = 0.32).

The wound complications rate was 12% in group A (three superficial, one deep of 34 fractures) and 10% in group B (one superficial, one deep of 20 fractures), but the difference was not statistically significant (χ^2^ = 0.4, *p* = 0.60) (Table 4). The duration of intervention was different between the two groups, and it was significantly longer in patients treated after 21 days (Table 4).

## 4. Discussion

The optimal management of closed displaced intra-articular calcaneal fractures remains controversial. Although some studies reported better outcomes after ORIF of displaced intra-articular calcaneal fracture [20,21,22,23,24], other RCTs showed that surgical treatment did not improve outcome when compared with non-operative treatment [25,26]. Furthermore, an increase in complications such as wound breakdown, infections, and reoperations have been reported after surgical treatment. A major goal of surgery is to achieve accurate anatomical reduction and a low complication rate [27]. The STA has been widely used to reduce soft tissue damage and postoperative complications. The STA allows a safer surgical exposure than the extended lateral approach (ELA) in terms of soft tissues management, and allows anatomical reduction of the fracture [28]. Moreover, the STA is useful for the treatment of closed DIACFs in patients with serious medical conditions requiring delayed surgical treatment [17,29]. Nosewicz et al., in a systematic review and meta-analysis, evaluated wound healing complications following STA compared to the ELA [30]. Nine studies were included, with 326 patients (331 fractures) treated by STA and 383 patients (390 fractures) treated by ELA. Ninety-nine per cent were Sanders type II/III fractures. The authors observed significantly fewer wound healing complications in the STA compared to ELA group. In particular, 11/331 (4.9%) and 82/390 (24.9%) wound healing complications were found in the STA and ELA groups, respectively. Holmes evaluated displaced intra-articular calcaneal fractures treatment by the STA with screw fixation [29]. No wound dehiscence, osteomyelitis, or deep surgical wound infection were found.

Many patients who suffered a major trauma and were referred to our trauma center were not candidates for early ORIF because of their critical or unstable medical conditions, and delayed definitive surgery was planned. Therefore, the present study assessed the complication rate and the quality of fracture reduction in a cohort of patients with Sanders type II and III DIACFs treated more than 21 days after the index trauma (group B), comparing them with patients treated within 21 days after injury (group A). To the best of our knowledge, this is the first study that compared the treatment of DIACFs using the STA before and after 21 days from the initial trauma.

The quality of reduction was not negatively influenced by the timing of definitive surgery. No statistical differences between group A and B were found in wound complication rate.

Only one study compared the incidence of wound complications of the STA considering the timing of surgery [31]. The authors concluded that delaying definitive fixation of closed DIACFs longer than two weeks increased wound complication rates when using the STA. The authors detected that the rate of wound complication raised with time because of the fragments’ consolidation and the soft tissue contracture. On the contrary, our findings suggest that prolonged waiting (beyond 3 weeks) is not detrimental for soft tissues using the STA, and did not result in a significantly increased wound complication rate, which is in line with results described in the literature. The different results could be explained considering that we included type II and III fractures according to Sanders classification, whereas Kwon et al. involved also Type IV fractures. Furthermore, the authors did not mention which type of hardware was used. We used specialized low-profile plates designed for the STA. Moreover, the ESTA was used in all our patients to minimize soft tissue manipulation caused by initial bone callus formation that necessitated more complex maneuvers for disimpaction and reduction. Li et al. assessed the optimal timing of surgery to achieve the lowest incidence of wound complications in a cohort of 53 patients [32]. According to the time interval between trauma to surgery, patients were classified into four groups: immediate (0–3 days), early (4–6 days), intermediate (7–14 days), and late (14–18 days). Deep infections and wound necrosis were found in 2 of 8 fractures (25%) of the immediate surgery group, needing surgical debridement and flap reconstruction. In the early group, 1 of 15 fractures developed superficial infection (6.7%) that healed with wound dressing and antibiotics. In the intermediate and late groups, all wounds healed uneventfully. The authors discouraged early surgery given the higher rate of wound complications.

The duration of surgery was greater in patients treated after 21 days or later given the presence of partial consolidation of the fracture, resulting in more difficult reduction. However, it did not result in a significantly poorer reduction quality in Group B compared with Group A.

The main limitations of the present study are the small sample size and the relatively short follow-up.

## 5. Conclusions

Delayed surgery does not negatively influence surgical results of ORIF of displaced intraarticular calcaneal fractures. It possible to delay surgical treatment of these injuries in patients who need stabilization following a major trauma. Sanders II and III DIACFs were adequately visualized using the STA to achieve anatomic reduction and stable fixation. No significant differences were found between two groups in wound complication rate. Further studies to verify the validity of these results are required.

## Figures and Tables

**Figure 1 jcm-12-02039-f001:**
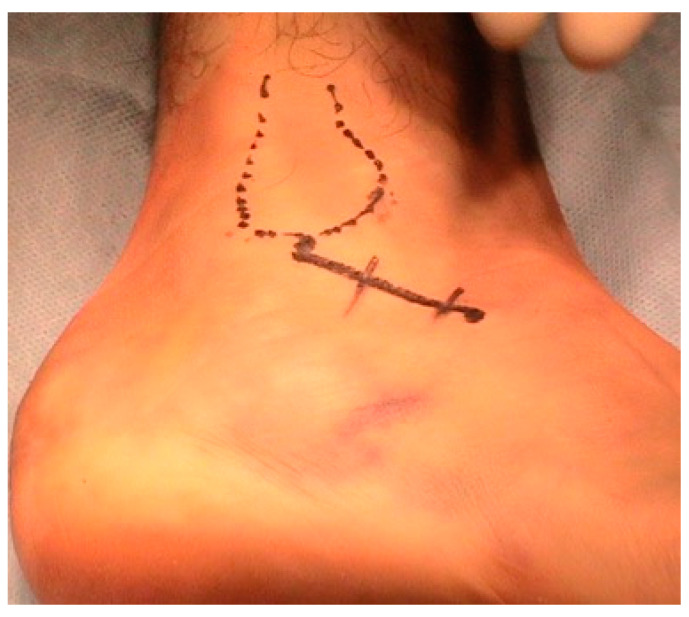
Surgical approach: A skin incision is made from the tip of the lateral malleolus to the base of the fourth metatarsal.

**Figure 2 jcm-12-02039-f002:**
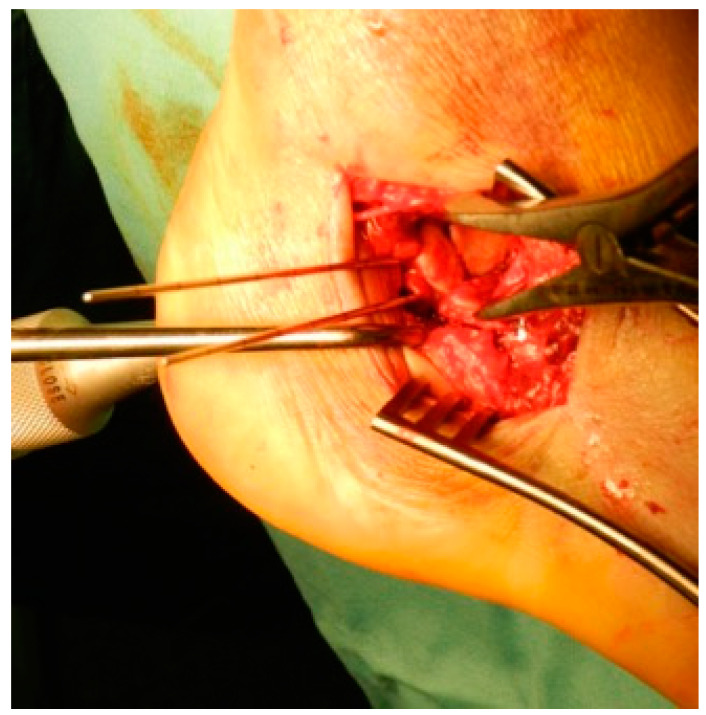
Reduction of the fragments with provisional fixation with K-wires.

**Figure 3 jcm-12-02039-f003:**
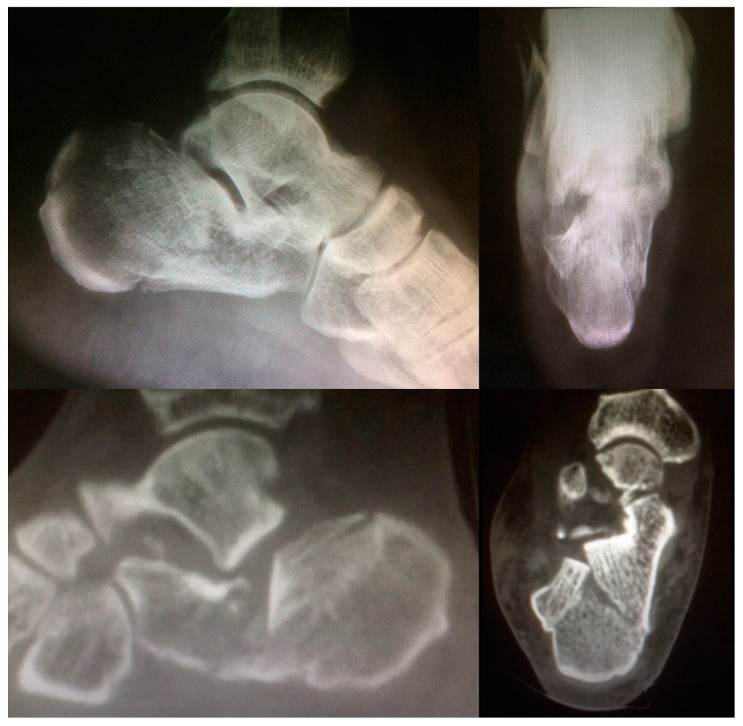
Preoperative X-ray and CT images of a Sanders Type III calcaneal fracture.

**Figure 4 jcm-12-02039-f004:**
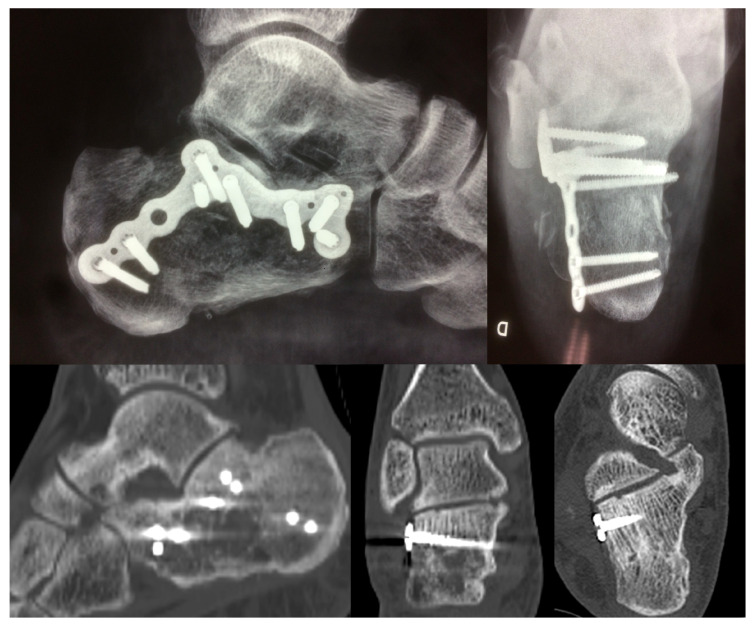
Postoperative X-ray and CT images.

**Table 1 jcm-12-02039-t001:** The Sanders classification of intra-articular fractures of the calcaneus is based on the number and location of the intra-articular fractures using semicoronal CT images. Fracture lines A, B, and C describe the position of the primary fracture line in relation to the posterior facet and the subtalar joint.

Sanders Type	Patterns of fracture		
Type I	Not displaced or minimally displaced		
		Type II A	A—Lateral third
Type II	Two-part fracture	Type II B	B—Central third
		Type II C	C—Medial Third
		Type II AB	AB—Involvement of the lateral and central aspects of the posterior facet of the calcaneus
Type III	Three-part fracture	Type II BC	BC—Involvement of the central and medial aspects of the posterior facet of the calcaneus
		Type II AC	AC—Involvement of the lateral and medial aspects of the posterior facet of the calcaneus.
Type IV	Four or more fragments		

**Table 2 jcm-12-02039-t002:** Surgical procedure step-by-step.

Patient Position	Lateral Decubitus
Intra-operatory aids	C-arm fluoroscopy
Anatomical markers	Tip of the lateral malleolus Base of the 4th metatarsal bone
Fragments exposure	Release of the inferior peroneal retinaculum Release of the CFL and subtalar ligaments SJ exposure
Steps for fragments reduction	Reduction of the PT fragment to the sustentacular constant fragment Reduction of articular fragments from posterior to anterior
Osteosynthesis	Temporary fixation with K-wires IF with plate and screws
Wound closure	

CFL: calcaneofibular ligament; SJ: subtalar joint; PT: Posterior Tuberosity; IF: Internal Fixation.

**Table 3 jcm-12-02039-t003:** Baseline characteristic of the sample.

Gender - Males (N = 32) - Females (N = 22)	
Mean Age sample, years 38.35 ± 11.77 SD	
Body Mass Index - Group A: 24.84 ± 3.2 - Group B: 25.74 ± 3.4	*p* = 0.34
Sanders Classification - II: N = 13 Group A, N = 8 Group B - III: N = 21 Group A, N = 12 Group B	
N = Number, SD = Standard Deviation	

**Table 4 jcm-12-02039-t004:** Results.

Wound Complications			
Group A	N. 4 (34 fractures; 12%)	3 superficial; 1 deep	*p* = 0.60
Group B	N.2 (20 fractures; 10%)	1 superficial; 1 deep	
**Duration of surgery (Min)**			
Group A	95.67 ± 22.93		*p* < 0.001
Group B	167.35 ± 30.93		

## Data Availability

All data generated or analyzed during this study are included in this published article. The datasets used and/or analyzed during the current study are also available from the corresponding author on reasonable request.

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
