# Peer review of "Delayed Surgical Treatment of Displaced Intra-Articular Calcaneal Fractures in Major Trauma Is Safe and Effective"

_jcm, 2023, doi:10.3390/jcm12052039_

Round 1

Reviewer 1 Report

The manuscript is interesting and well-written. However, I have several comments for the authors.

Lines 53-54: “Open reduction and internal fixation (ORIF) with the extensile lateral approach (ELA) is considered the gold standard treatment of these fractures. This approach provides excellent exposure of the posterior facet and lateral wall.” References should be provided.

Lines 56-57: “Minimal incisions, including the sinus tarsi  approach (STA), have been introduced to reduce the risk of wound complications.” References should be provided.

The authors divided patients into two groups: Group A: patients treated within 21 days after injury;

Group B: patients treated more than 21 days after injury. Why were 21 days selected as cut off point? Is there a particular reason?

In the abstract it is reported that the authors included polytrauma patients but there is no mention about polytrauma patients in section 2.1 of the methods.

Table 3: the authors reported the baseline characteristics of the entire cohort.  The authors compared only BMI results between the two groups. What about the other baseline characteristics? Are there differences?

Could the authors add a clinical case (with X-rays and CT)?

Ethical committee approval number and date should be provided.

Author Response

Lines 53-54: “Open reduction and internal fixation (ORIF) with the extensile lateral approach (ELA) is considered the gold standard treatment of these fractures. This approach provides excellent exposure of the posterior facet and lateral wall.” References should be provided.

Authors: we add references:

  • Wallace B, Chhabra A, Narayanan A, O'Neill D, Xi Y, Van Pelt M, Wukich DK, Liu G, Sanders D, Lalli T. Low Risk of Wound Complications With Sinus Tarsi Approach for Treatment of Calcaneus Fractures. J Foot Ankle Surg. 2022 Jul-Aug;61(4):771-775.
  • Peng C, Yuan B, Guo W, Li N, Tian H. Extensile lateral versus sinus tarsi approach for calcaneal fractures: A meta-analysis. Medicine (Baltimore). 2021 Aug 6;100(31):e26717. doi: 10.1097/MD.0000000000026717.

Lines 56-57: “Minimal incisions, including the sinus tarsi  approach (STA), have been introduced to reduce the risk of wound complications.” References should be provided.

Authors:

  • reference: Wallace B, Chhabra A, Narayanan A, O'Neill D, Xi Y, Van Pelt M, Wukich DK, Liu G, Sanders D, Lalli T. Low Risk of Wound Complications With Sinus Tarsi Approach for Treatment of Calcaneus Fractures. J Foot Ankle Surg. 2022 Jul-Aug;61(4):771-775.

The authors divided patients into two groups: Group A: patients treated within 21 days after injury;

Group B: patients treated more than 21 days after injury. Why were 21 days selected as cut off point? Is there a particular reason?

Authors: We chose 21 days as cut off point because of the fibrous callus formation which . Ref. Weatherford BM, Matz J, Kandemir U. Displaced Intra-articular Calcaneus Fractures: Extensile Lateral and Less Invasive Approaches. Instr Course Lect. 2023;72:543-554.

In the abstract it is reported that the authors included polytrauma patients but there is no mention about polytrauma patients in section 2.1 of the methods.

Authors: we included patients who were hospitalized for major trauma. Most of these patients are not candidate to early ORIF, and a delaied surgery is required. Many have been submitted to multiple surgery (Orthopaedic damage control, abdominal surgery, abdominal packing and depacking). All the patients treated within 21 days suffered an isolated calcaneal fracture. However, the aim of the study is not to report the results of the treament between patient affected by isolated calcaneal fractures and major trauma, but the quality of reduction and complication between early and delayed treatment. Therfore, we don’t think considrable introduce this variable in the materials and methods section. Furthermore, we did not report the conomitant fractures, comorbilitis, and other surgeries for the same reason (See tab III).

During the discussion, we explain that the main reason for delayed surgey is a major trauma.

Table 3: the authors reported the baseline characteristics of the entire cohort.  The authors compared only BMI results between the two groups. What about the other baseline characteristics? Are there differences?

Authors: See previous comment.

Could the authors add a clinical case (with X-rays and CT)?

Authors: we included a clinical case (Figure 3 and 4).

Ethical committee approval number and date should be provided.

Authors: we add the information about the Etical Committee approval -(Line 85).

Thank you for reviewed our manuscript, and gave us the opportunity to improve it.

kind regards

Reviewer 2 Report

Compliments to the authors for the very interesting idea at the basis of their study. For the case history reported and the quality of the manuscript, it only deserves a minor revision.

1. If possible, the authors should include radiographic documentation of one patient from the case history (would increase the quality of their work).

2. It should be emphasised that polytrauma and the presence of associated psychiatric comorbidities are important negative prognostic factors in this type of patient (if useful, doi: 10.3390/jcm11195660)

3. Line 141, the authors mention (Figure 2 A-B) which seems to be missing in the manuscript, please check.

4. On line 177, along with citation 22, if the authors consider it appropriate they could emphasise how important it is to also avoid an often misunderstood complication such as checkrein deformity (doi: 10.3390/medicine58081072)

Author Response

  1. If possible, the authors should include radiographic documentation of one patient from the case history (would increase the quality of their work).

Authors: we included fig 3 and 4

  1. It should be emphasised that polytrauma and the presence of associated psychiatric comorbidities are important negative prognostic factors in this type of patient (if useful, doi: 10.3390/jcm11195660)

Authors: we add reference n.12

  1. Line 141, the authors mention (Figure 2 A-B) which seems to be missing in the manuscript, please check.

Authors: we delate fig 2 A-B and corret into the text (line 141)

  1. On line 177, along with citation 22, if the authors consider it appropriate they could emphasise how important it is to also avoid an often misunderstood complication such as checkrein deformity (doi: 10.3390/medicine58081072).

Authors: We add ref n.24